# Membraneless water filtration using $CO_2$

Sangwoo Shin[1,*,†], Orest Shardt[1,*,†], Patrick B. Warren[2] & Howard A. Stone[1]

Water purification technologies such as microfiltration/ultrafiltration and reverse osmosis utilize porous membranes to remove suspended particles and solutes. These membranes, however, cause many drawbacks such as a high pumping cost and a need for periodic replacement due to fouling. Here we show an alternative membraneless method for separating suspended particles by exposing the colloidal suspension to $CO_2$. Dissolution of $CO_2$ into the suspension creates solute gradients that drive phoretic motion of particles. Due to the large diffusion potential generated by the dissociation of carbonic acid, colloidal particles move either away from or towards the gas–liquid interface depending on their surface charge. Using the directed motion of particles induced by exposure to $CO_2$, we demonstrate a scalable, continuous flow, membraneless particle filtration process that exhibits low energy consumption, three orders of magnitude lower than conventional microfiltration/ultrafiltration processes, and is essentially free from fouling.

[1] Department of Mechanical and Aerospace Engineering, Princeton University, Princeton, New Jersey 08544, USA. [2] Unilever R&D Port Sunlight, Bebington, Wirral CH63 3JW, UK. * These authors contributed equally to this work. † Present addresses: Department of Mechanical Engineering, University of Hawaii at Manoa, Honolulu, Hawaii 96822, USA (S.S.); Bernal Institute and School of Engineering, University of Limerick, Castletroy, Limerick V94 T9PX, Ireland (O.S.). Correspondence and requests for materials should be addressed to H.A.S. (email: hastone@princeton.edu).

With global demand for clean water increasing, there is a continuing need to improve the performance of water treatment processes[1,2]. Membraneless separation of suspended particles is conventionally achieved by sedimentation, which relies on the gravitational force[3]. When particle suspensions are stable, for example, due to the particles being small, neutrally buoyant and highly charged, unassisted sedimentation is ineffective and additives are needed to induce aggregation (coagulation or flocculation[4]) and accelerate sedimentation. Therefore, inducing directed motion of colloidal particles in stable suspensions is critical to membraneless separation processes.

There are several ways to induce directed motion of colloidal particles such as using an external force, for example, electrostatic[5], dielectric[6], magnetic[7], acoustic[8], or optical[9], inertial effects[10] and so on. One less known but significant driving force is a chemical gradient, which causes diffusiophoresis[11]. Typically, diffusiophoresis is observed in liquid environments where chemical gradients are generated by contact between solutions with different solute concentrations[12–15].

Here we show that diffusiophoresis of colloidal particles can be achieved by dissolution of a gas into a liquid, namely $CO_2$ dissolution into water and its subsequent dissociation that produces ion concentration gradients. Using this principle, we show that the directed motion driven by dissolution of $CO_2$ can be exploited to separate particles with very low energy consumption. Moreover, $CO_2$ is abundant, biologically benign when dissolved in water, and can be easily separated. These features make $CO_2$ dissolution a very promising means to clean water especially for the developing world, which requires water purification technologies that have low energy demands and are not chemically intense[2].

## Results

**Gas-induced diffusiophoresis.** Dissolved $CO_2$ establishes an equilibrium with water through the overall reaction[16,17]

$$CO_2 + H_2O \underset{k_b}{\overset{k_f}{\rightleftharpoons}} H^+ + HCO_3^- \qquad (1)$$

where $k_f$ and $k_b$ are the forward and backward reaction rate constants[18], respectively. Reaction (1) implies that transient ion gradients can be created in aqueous particle suspensions when they are exposed to $CO_2$ gas. Most significantly, the generated ions exhibit a large difference in their diffusivities[19] ($D_{H^+} = 9.3 \times 10^{-9} \, m^2 \, s^{-1}$ and $D_{HCO_3^-} = 1.2 \times 10^{-9} \, m^2 \, s^{-1}$), which may lead to strong particle diffusiophoresis due to a large diffusion potential ($\approx 92 \, mV$ for a 100-fold concentration difference, whereas in ordinary salt gradients, the potential is $25 \, mV$ for $Na^+/Cl^-$ and $0.1 \, mV$ for $K^+/Cl^-$).

To test the hypothesis that exposure to $CO_2$ can induce sufficient ion concentration gradients to drive particle motion, we developed a microfluidic system in which we create a stable gas–liquid interface (Fig. 1a, see Methods section). The microfluidic device, which is made out of ultraviolet-curable epoxy to prevent permeation of gases[20], has two main channels bridged by multiple narrow pores that contain a colloidal suspension (Fig. 1b,c).

Upon exposure to $CO_2$ gas (pressure $p_{CO_2} = 136 \, kPa$), negatively charged particles (polystyrene, diameter $\approx 0.5 \, \mu m$, zeta potential $\approx -70 \, mV$) immediately migrate away from the two gas–liquid interfaces (Fig. 1e,g, Supplementary Movie 1). In contrast (Fig. 1d, Supplementary Movie 2), no appreciable motion of these particles, other than Brownian motion, is observed upon exposure to $N_2$ gas ($p_{N_2} = 136 \, kPa$). Based on our hypothesis, the observed particle migration is mainly induced by the diffusion potential due to $H^+$ and $HCO_3^-$. This assertion is supported by experiments with positively charged particles (amine-functionalized polystyrene, diameter $\approx 1 \, \mu m$, zeta potential $\approx 60 \, \mu m$) that migrate toward the gas–liquid interfaces upon exposure to $CO_2$, forming clouds of particles near these interfaces (Fig. 1f,h, Supplementary Movie 3).

The presence of ion concentration gradients, responsible for driving the motion of particles, was confirmed by observing the response of liquid columns filled with pH indicator solution to gas exposure. We used Oregon Green fluorescent dye (0.1 mM) as the pH indicator due to its linear dependence of fluorescence

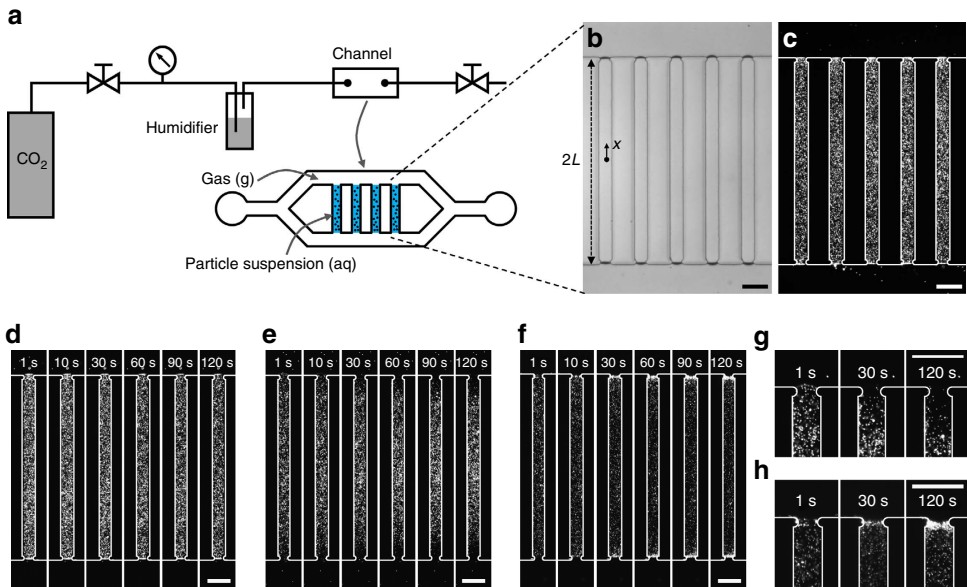

**Figure 1 | Motion of colloidal particles upon exposure of aqueous suspensions to gas.** (**a**) Experimental set-up for creating a liquid column with a stable gas–liquid interface. (**b**) Bright-field and (**c**) corresponding fluorescence microscopy images of suspension-filled channels exposed to gas, where the length of the channel is $2L = 800 \, \mu m$. The width and the height of the channels are 60 and 40 $\mu m$, respectively. The ends of the pores are slightly contracted to effectively fix the liquid columns. (**d**) Image sequence showing particles (polystyrene, diameter 0.5 $\mu m$) exposed to $N_2$. (**e,f**) Image sequences showing directed motion of particles with (**e**) negative (polystyrene, diameter 0.5 $\mu m$) and (**f**) positive (amine-functionalized polystyrene, diameter 1 $\mu m$) surface charge exposed to $CO_2$. (**g,h**) Close-up images of the time evolution of the (**g**) negatively charged and (**h**) positively charged particles near the gas–liquid interface during exposure to $CO_2$. The gas pressures in all cases are 136 kPa. All scale bars are 100 $\mu m$.

 

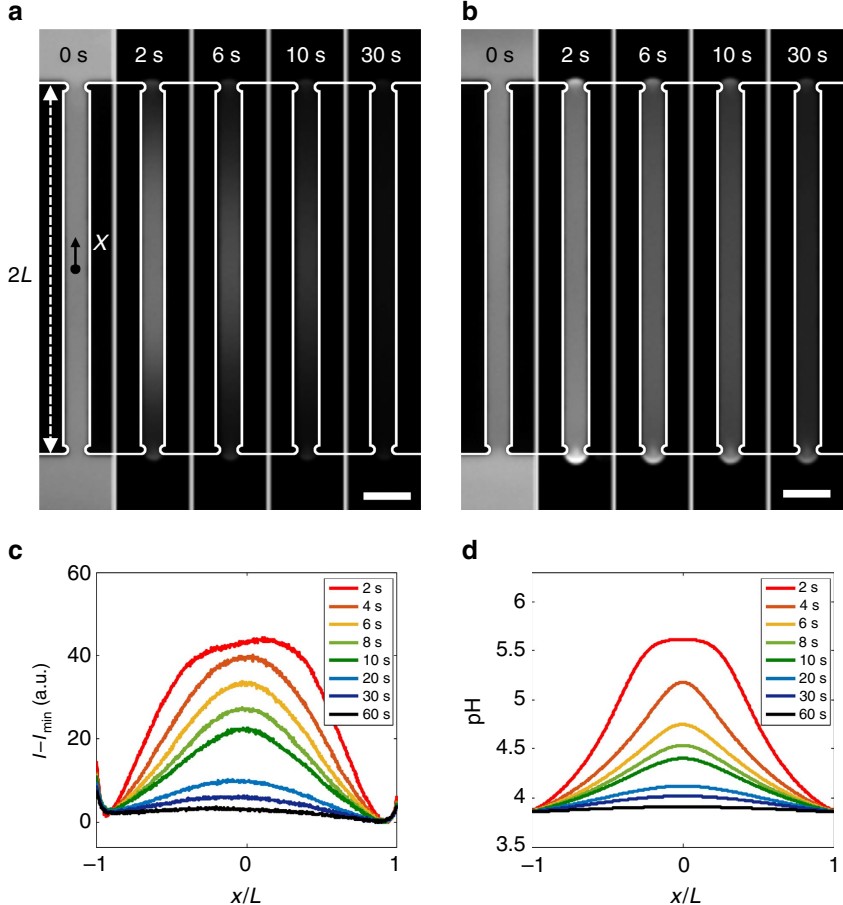

**Figure 2 | pH change in water upon exposure to gas.** (**a**,**b**) Image sequences showing fluorescence intensity in aqueous Oregon Green solution (0.1 mM) exposed to (**a**) $CO_2$ and (**b**) $N_2$, both at a pressure of 136 kPa. (**c**) Fluorescence intensity ($I$) distribution, reported as $I - I_{min}$ to account for photobleaching, along the centre of the channel shown in **a**. $I_{min}$ is the minimum intensity in the channel. (**d**) Computed pH distribution in water exposed to $CO_2$ at 136 kPa. All scale bars are 100 μm.

intensity on pH over a broad pH range[21]. As the water is exposed to $CO_2$ at $p_{CO_2} = 136$ kPa, the fluorescence intensity immediately decreases near the gas–liquid interfaces, and then the decreasing intensity propagates toward the middle of the column, indicating that pH decreases from the interfaces inward (Fig. 2a). The high intensity evident near the gas–liquid interface is due to the curvature of the meniscus. In contrast to $CO_2$, exposure to $N_2$ at the same pressure does not produce spatial variations in fluorescence intensity, but only a gradual decrease of fluorescence intensity due to photobleaching (Fig. 2b).

Solving coupled diffusion-reaction equations predicts the transient pH distribution under exposure to $CO_2$. The dissolved $CO_2$ and ion concentrations are coupled through the net forward rate of reaction (1) $r = k_f c_c - k_b c_i^2$, where $c_c(x,t)$ is the concentration of dissolved $CO_2$, and $c_i(x,t)$ gives the concentrations of both generated ions, $H^+$ and $HCO_3^-$, which are effectively equal due to local charge neutrality[22,23]. Then, solving the diffusion-reaction equations $\partial c_c/\partial t = D_c \nabla^2 c_c - r$ and $\partial c_i/\partial t = D_i \nabla^2 c_i + r$ yields the transient ion concentration distribution, where $D_c$ is the diffusivity of $CO_2$ and $D_i$ is the ambipolar diffusivity of the ions. The calculated pH distribution (Fig. 2d) shows reasonable agreement with the observed fluorescence intensity profile (Fig. 2c). As the pH decreases from 5.6 (the pH of water in equilibrium with the atmospheric $CO_2$ concentration, $p_{CO_2} = 40$ kPa) down to 3.8 for $p_{CO_2} = 136$ kPa, a nearly 100-fold difference in ion concentrations occurs within the column.

This chemical gradient generated by dissolution of $CO_2$ induces particle migration via diffusiophoresis. The particle diffusiophoretic velocity is expressed as $\mathbf{u}_p = \Gamma_p \mathbf{\nabla} \ln c_i$, where the prefactor $\Gamma_p$ is the diffusiophoretic mobility, which is a measure of the strength of diffusiophoresis based on surface-solute interactions[11]. By solving an additional advection–diffusion equation for the particles, we obtain the particle dynamics driven by the dissolution of $CO_2$ (Supplementary Discussion). The one-dimensional transport model captures the key dynamics that are consistent with the experimental observations for both the negatively (Fig. 3a,c) and positively charged particles (Fig. 3b,d), where the negatively charged particles move away from the gas–liquid interface and subsequently focus to the centre of the column; motion is in the opposite direction for positively charged particles. From calculations at other conditions than those of these experiments, we note that the minimum $CO_2$ pressure needed for inducing appreciable particle motion is much lower than 136 kPa, but it must be sufficiently higher than the atmospheric partial pressure of $CO_2$ (see Supplementary Fig. 1, Supplementary Table 2, and Supplementary Discussion).

**Membraneless water filtration.** The observed colloid dynamics suggest an alternate route to separating particles without the use of membranes or filters, which is by exposing the colloidal suspension to $CO_2$ to concentrate the particles locally as in field-flow fractionation[24,25]. A continuous flow particle filtration

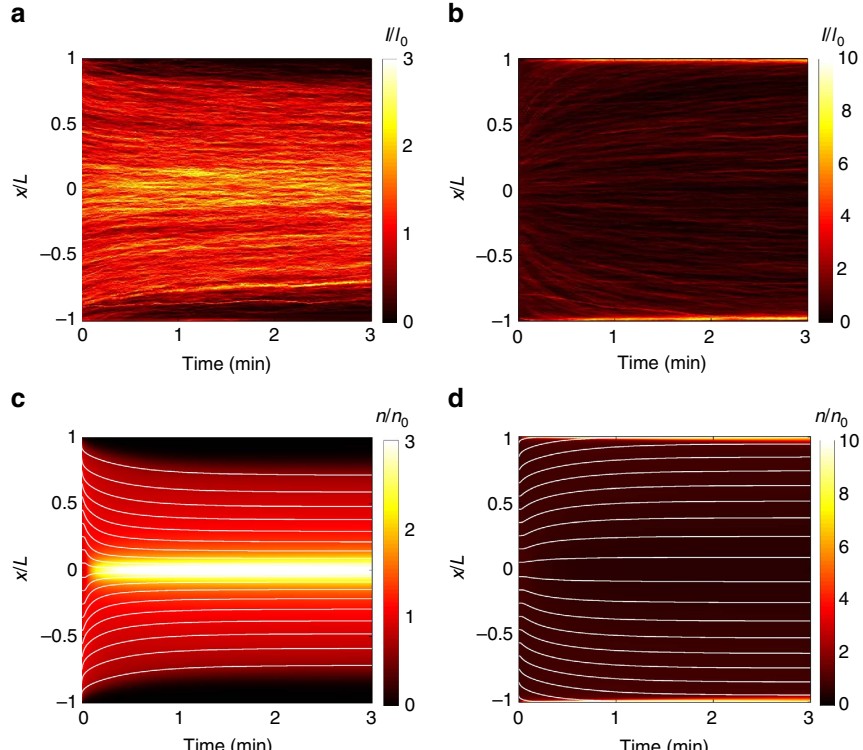

**Figure 3 | Spatio-temporal evolution of particle distributions upon exposure to CO₂.** (**a,b**) Experimental and (**c,d**) computed plots for (**a,c**) negatively charged (polystyrene, diameter 0.5 μm) and (**b,d**) positively charged particles (amine-functionalized polystyrene, diameter 1 μm) exposed to $CO_2$ at 136 kPa. $I$ and $n$ represent fluorescence intensity and particle concentration, respectively. Subscript 0 indicates a quantity at $t = 0$ s. The experimental plots are obtained by averaging the intensity over the width of the channel. The white curves in (**c,d**) indicate the trajectories of individual particles obtained by integrating their diffusiophoretic velocities and neglecting Brownian motion. The computations for (**d**) account for the presence of a low concentration of ionic solutes in the positive particle suspensions (see Supplementary Fig. 2 and Supplementary Discussion).

device is illustrated in Fig. 4a. A colloidal suspension flows through a straight channel in a gas permeable material, in this case, polydimethylsiloxane (PDMS)[26,27]. Separated by a thin wall, a $CO_2$ gas channel passes parallel to the main flow channel. The gas permeates through the wall and dissolves into the flow stream, inducing particle motion transverse to the flow direction; the direction of motion depends on the particles' surface charge. An air channel on the other side of the wall prevents saturation of $CO_2$ in the suspension, thus sustaining a greater concentration gradient and enabling further migration of particles by diffusiophoresis. After all of the particles have been concentrated at one side of the channel, they are separated by splitting the flow near the outlet.

The removal of colloidal particles using our method is presented in Fig. 4b–e, where the device is made out of standard PDMS. Negatively charged polystyrene particles, which are initially distributed uniformly (Fig. 4b), migrate away from the wall that is adjacent to the $CO_2$ channel as they flow downstream, and are eventually separated from the main stream (Fig. 4c). Positively charged particles can also be filtered using the same principle (Fig. 4d,e). The particle removal rate in terms of log reduction value is estimated to be 5.3 (see Methods and Supplementary Fig. 3), which is comparable to values for conventional microfiltration and ultrafiltration techniques[28].

Since the demonstrated method does not involve any filters, its energy consumption is low compared to conventional filtration methods. Energy dissipation is reduced because the fluid may flow through a channel that is much wider than the particles instead of passing through pores that are smaller than the particles. Based on the current channel geometry (width × height × length = $0.1 \times 0.02 \times 30 \, mm^3$) and flow rate

($2 \, \mu l \, h^{-1}$), the pressure drop across the channel is $\Delta p \approx 0.2 \, kPa$. Assuming 50% recovery of water from the suspension, the energy consumption is estimated as $0.12 \, mW \, h \, l^{-1}$, at least three orders of magnitude lower than the energy consumed in typical microfiltration and ultrafiltration processes[29,30] ($0.1 \sim 20 \, W \, h \, l^{-1}$). The energy efficiency can be further enhanced by optimizing the channel geometry such as increasing the channel height to reduce the pressure drop for a given flow speed.

Furthermore, the proposed device is easily scalable due to its simple design, for example by using parallel flow channels (Fig. 4f). Such a design allows easy scaling-up in both the in-plane and out-of-plane directions due to the isotropic permeability of gas in PDMS[31]. As a proof of concept, a 10-fold upscaled device made out of a monolithic PDMS block is shown in Fig. 4g–j. Likewise, a million-fold upscaling, which is feasible using standard microfabrication techniques, would purify water at ≈50 l per day. The dissolved $CO_2$ in the output streams can be removed by equilibration with air (see for example, ref. 27). The waste stream could pass then through the same process multiple times to further concentrate the particle stream and extract additional water.

## Discussion

By exploiting the dissolution of $CO_2$ as a way to drive particle motion, we have demonstrated a membraneless method that is capable of separating the types of particles that would otherwise form a stable suspension. Due to their small size, surface charge, and the absence of dissolved ions to screen the charge (that is, a large Debye length), the particles we separate would be difficult to remove via sedimentation. The proposed technique is easily scalable and shows a three order of magnitude lower energy

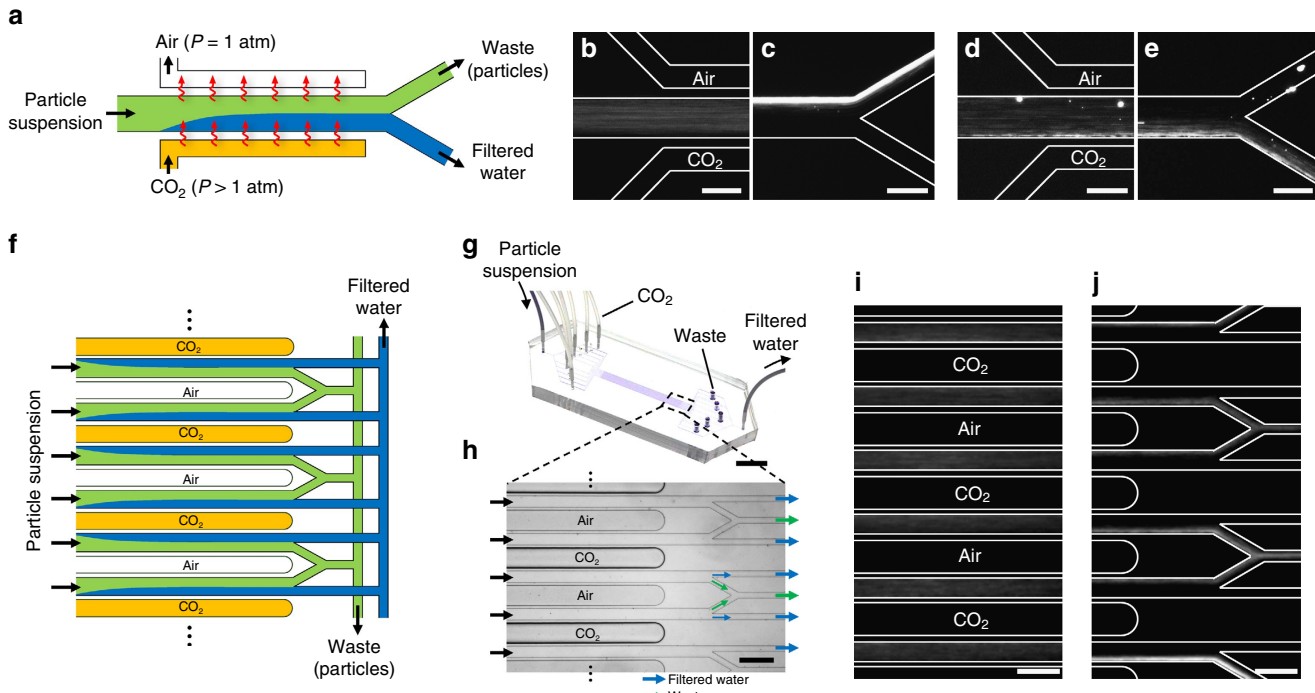

**Figure 4 | Continuous-flow membraneless water filtration using dissolution of $CO_2$.** (**a**) Schematic of the water filtration process. The channel material (PDMS) allows permeation of $CO_2$ gas into the particle suspension, inducing directed particle motion normal to the flow direction. (**b–e**) Fluorescence microscopy images of (**b,c**) negatively and (**d,e**) positively charged particles migrating transverse to the flow direction. Images are taken near the (**b,d**) entrance and (**c,e**) end of the flow channel, where the channel length is 3 cm. PDMS walls (30 µm wide) separate the parallel gas and flow channels. The $CO_2$ channel is kept at a pressure of 170 kPa whereas the air channel is left open to the atmosphere. The air channel prevents saturation of $CO_2$ in the suspension, thus enhancing particle migration transverse to the flow direction. (**f**) Schematic of a scalable water filtration process based on the unit device shown in **a**. (**g**) Photograph of water filtration device having 10 parallel flow channels in a monolithic PDMS block. (**h**) Bright-field microscopy image of the microfluidic channels near the outlets. The $CO_2$ channels are kept at 136 kPa whereas the air channels are left open to the atmosphere. (**i,j**) Fluorescence microscopy images of the negatively charged particles near the (**i**) entrance and (**j**) end of the flow channel. The scale bars are (**b–e**) 100 µm, (**g**) 1 cm and (**h–j**) 200 µm.

consumption than conventional filtration systems, suggesting the technique could be particularly beneficial in both the developing and developed worlds[2]. We suggest that the proposed technique could be used either to replace microfiltration and ultrafiltration or benefit such conventional filtration processes by mitigating membrane fouling. Also, considering that most bioparticles are charged[32], the proposed method can be utilized to remove bacteria and viruses without chlorination or ultraviolet treatment[2,30].

Efficient scaling-up of the process to significant flow rates requires optimization of the channel dimensions, the flow rate in each channel, the split ratio, and the number of parallel channels. While frictional losses can be reduced by using a wider or taller channel, increasing the width decreases the magnitude of the concentration gradient. Reduced gradients and lower residence times due to higher flow rates can be compensated by using longer channels at the expense of increasing the required energy input. Longer and narrower channels can also be used to overcome reductions in the diffusiophoretic driving force in applications with sufficient background salt concentrations. To assess the extent of the decrease in the diffusiophoretic driving force with increasing salt concentrations, further experiments with controlled amounts of common salts as well as representative samples with typical mineral contents from industrial and natural water sources would be valuable. Long-term experiments are also needed to assess fouling, such as by proteins or bacterial extracellular polymeric substances.

Our demonstration of particle transport driven by $CO_2$ dissolution is not limited to particle separation, but may also explain the mechanisms underlying some applications that involve dissolution of $CO_2$, such as the stabilization of bubbles via particle assembly at their surfaces[33] and the enhancement of froth flotation through the use of $CO_2$ bubbles[34]. These applications require collection of particles at gas–liquid interfaces, and Fig. 1h demonstrates that even hydrophilic particles can be collected at a gas–liquid interface from a dilute suspension.

## Methods

**Liquid column experiments.** The microfluidic device for the liquid column experiments shown in Fig. 1 was made out of ultraviolet-curable epoxy (NOA81, Norland Products) using the microfluidic sticker technique[35]. The channels were 40 µm thick. Humidified gas is fed through the microfluidic channel, which is initially filled with colloidal suspension. The gas pushes out all the suspension in the main channels, leaving behind columns of colloid in the pores that bridge the two main channels, thus creating a stable gas–liquid interface. This is facilitated by small constrictions at the ends of the pores which serve to break the liquid columns at the right place. Once the suspension is flushed out, the outlet is closed to stop the gas flow, which may otherwise interrupt the particle motion. The gas pressure is monitored and controlled by a pressure regulator (Type-10, Bellofram) equipped with a digital pressure gauge (DPG1000B, Omega). The gas pressure of 136 kPa (5 psi above atmospheric pressure) was chosen to allow flow through the device to an outlet at atmospheric pressure. The colloidal particles (polystyrene, Bangs Laboratories; amine-functionalized polystyrene, Sigma-Aldrich) were prepared by dilution in deionized water (Milli-Q, Millipore) to a volume fraction of 0.01%. pH visualization experiments were conducted in the same manner. The pH indicator was prepared by diluting fluorescent dye (Oregon Green 488, Thermo Fisher Scientific) in deionized water at 0.1 mM. The colloidal particles and the dye were observed with an inverted fluorescence microscope (DMI4000B, Leica). The zeta potentials of the colloidal particles were measured using Zetasizer Nano-ZS (Malvern Instruments).

**Water filtration experiments.** The microfluidic device for the water filtration experiments shown in Fig. 4 was made out of PDMS (Sylgard 184 Elastomer Kit, Dow Corning) using a conventional soft lithography technique. The monomer and the cross linker were mixed at a weight ratio of 10:1. A syringe pump (PhD Ultra, Harvard Apparatus) was used to flow the colloidal suspensions through the filtration device. For the positively charged particle experiments, in order to prevent adhesion of amine-functionalized polystyrene particles to the channel walls, the PDMS was modified with amine by flowing a 1 vol% aqueous solution of 3-aminopropyltrimethoxysilane (Sigma-Aldrich) through the PDMS channel for 20 min followed by rinsing with deionized water for 10 min prior to experiments[36].

**Log reduction value measurement.** Owing to the low flow rate $(2\,\mu l\,h^{-1})$ and high fluorescence intensity of the particles, we are able to count the number of individual particles passing through the filtrate stream (Supplementary Fig. 3). During 5 min, out of $\sim 2.2 \times 10^7$ total particles only 104 pass through the filtrate stream, which corresponds to a log reduction value of 5.3.

**Simulations.** The coupled reaction-diffusion equations for the $CO_2$ (Supplementary equation (2)) and ion concentrations (Supplementary equation (8)) together with the advection–diffusion equation (Supplementary equation (9)) for the particle concentration were solved in MATLAB using pdepe. Exploiting symmetry, solutions were computed for $x \in [0, L]$ using a fine grid with refinement at $x = 0$ and $x = L$ to resolve gradients in concentration. The particle trajectories $x_p(t)$ were computed by interpolating the pre-computed ion concentration solution and numerically integrating $dx_p/dt = u_{dp}(x_p(t), t)$ for several uniformly spaced initial positions using ode113 also in MATLAB. Additional details are provided in the Supplementary Discussion. The model parameters are listed in Supplementary Table 1.

**Data availability.** The data that support the findings of this study are available from the authors upon reasonable request.

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

## Acknowledgements

We acknowledge support from Unilever Research. We thank Bhargav Rallabandi and Suin Shim for valuable discussions. O.S. thanks the Natural Sciences and Engineering Research Council of Canada (NSERC) for a postdoctoral fellowship.

## Author contributions

All authors conceived the project. S.S. performed the experiments. O.S. performed the simulations. All authors discussed the results and wrote the paper.

## Additional information

**Competing interests:** P.B.W. discloses a substantive ($> \$10,000$) stock holding in Unilever PLC. The authors have filed a provisional patent application related to this work. The remaining authors declare no competing financial interests.

**Publisher's note**: 

