## [Peer Review File · Nature Communications]

Reviewers' comments:

Reviewer #1 (Remarks to the Author):

This is an interesting and well written contribution.

Microfluidic filtration of colloids has been addressed in a number of publications from Montpellier, Aachen and Twente. It is growing field as to study membrane filtration and colloidal filtration phenomena. (Linkhorst, J., Beckmann, T., Go, D., Kuehne, A. J. C., & Wessling, M. (2016). Microfluidic colloid filtration. *Scientific Reports*, 6, 22376–8. <http://doi.org/10.1038/srep22376>)

The use of CO₂ to establish pH gradients in PDMS based chips has been extensively studied, but not cited. de Jong, J., Verheijden, P. W., Lammertink, R. G. H., & Wessling, M. (2008). Generation of local concentration gradients by gas-liquid contacting. *Analytical Chemistry*, 80(9), 3190–3197. <http://doi.org/10.1021/ac7023602>

A membrane free separation process in a microfluidic device using pH gradients is novel and worth publishing. The visually observed separation seems very effective. Yet, it would be great to quantify the sharpness of the separation. In membrane filtration, retention curves are used. how long has the channel to be to have no particles left in the particle-free stream? How clean is the "filtrate"? Any thoughts how the LRV (log reduction value) can be ?

While the work is conceptually inspiring, it will have a hard time to be implemented. Anybody who has worked with these chips knows how difficult it is to get them operate properly. Filtering a real world solution will clog the device immediately.

Energy consumption is used as an argument why this device is superior of a membrane device. Energy consumption is not much of an issue in dilute streams. Short-term fouling control and long term drift though is. Energy consumption only becomes relevant in high fouling streams operating at high Re numbers due to recirculating up to 90% of the retentate. Yet, such streams can not be handled by this device.

Nonetheless, this paper should be published and the authors may change some of their wording and reflections if they are susceptible to the arguments and their own experience in chip operation.

Reviewer #2 (Remarks to the Author):

PART 1

The authors address the idea of membrane filtration using CO₂. The core idea is related to field flow fractionation (FFF, as Giddings described), in which particles are pulled toward or away from one region of the capillary cross section, in this case by diffusiophoresis, and then the output is split. The diffusiophoresis is caused by an electrolyte gradient of dissociated carbonic acid, which arises as the CO₂ permeates one side of the PDMS material and dissociates into H⁺ and HCO₃⁻. This manuscript is very interesting, and could be suitable for Nature Comm, if the authors can address the questions and comments below.

- questions. The scientific or engineering questions need to be clarified. Right now the manuscript is not driven by strong questions, but rather by a clever idea.

- zeta potentials. What is the zeta potential of the amine particles? The negative PSL particles (and what are the surface groups)? The PDMS (and if small, does this allow particle adhesion)? The zetas will vary as a function of pH; the authors have chosen the pH for pN₂ = 136 kPa. Diffusiophoresis depends upon the difference in the zeta potentials of the particles and the surface,

and so these are critical to provide in the manuscript.

- flow profile. If the zetaPDMS is not 0, then the walls will contribute to the flow profile within the capillary. Have the authors verified the flow profile by examining the particle speeds throughout the depth of the cell (~60 um). How does this affect the separation? This might be especially important when the surface is not PDMS, if other materials would be used in commercial applications?

- pN2. The authors should discuss briefly in the manuscript why they chose pN2 = 136 kPa. In the SI the authors mention that this is a pressure where the speed of the particles does not change much with pN2, if I understand correctly? They might explain this in a sentence or two, and offer a reason why this is a good criterion for the choice of pressure?

- fouling. On p 4 the authors say that the membrane filtration idea is "easily scalable". What is the role of fouling -- due to either particles or bacteria -- of the PDMS, especially over months? Would this or other challenges affect the scalability? Or the membrane's cleaning cycle or useful lifetime?

- ionic strength. Many application will require finite salt concentrations, perhaps of NaCl or other salt. DP transport is reduced at higher ionic strength. Will the mechanism still be effective, say at 1 mM, 10 mM, 150 mM? What will need to change?

- advantages. The authors mention a 1000x reduction in energy (p 4, paragraph 1), which is a clear advantage. Even a 10x reduction is huge. They should address this with a bit of additional explanation, since this reduction seems so big. It is also very useful that the particle transport depends primarily on the particle zeta potential, and less so the particle size. This can be used to advantage in many situations. Also, the gas transport can be very rapid, which would allow rapid changes in the ionic strength as needed. The authors might point out these additional advantages.

PART 2

Overall, I do not understand why this paper is suitable for a high impact journal such as Nature Communications. It shows a unique phenomenon and a simple demonstration but I think the overall utility is somewhat oversold.

The comparison of the scalability and the power consumption to current practical membranes is not accurate I think. Current membranes are relatively energy efficient and for simple situations such as those demonstrated in this paper can even be run under gravity. Also the recovery of such membranes is over 95% not 50%. The comparison and scale up from a small microfluidic device to a large industrialized system is not appropriate I think.

Also, membranes provide a barrier for particle filtration, an important consideration due to the need to remove microorganisms, something that the proposed system cannot assure.

Other than this, the work conducted is elegant and provides a nice insight into unusual situations where diffusiophoresis can be utilized.

Reviewer #3 (Remarks to the Author):

This manuscript entitled "Membraneless water filtration using CO₂" reports the directed motion of particles induced by exposure to CO₂. The authors demonstrated a scalable, continuous flow, membraneless particle filtration process. The reviewer thinks that the idea is noteworthy and very promising method to clean water and to realize low energy consumption. However, it is difficult to

make a judgment whether the water filtration using CO₂ really work for chemical or biological fouling. It is true that the particle migration observed in this manuscript is mainly induced by the diffusion potential due to H⁺ and HCO₃⁻. However, natural organic matter in water has usually been known as complex mixtures consisting of hydrophobic and hydrophilic materials with a broad range of molecular weights.

The reviewer thinks that this manuscript needs additional description for above and the following comments to publish in Nature Communications.

For the authors, comments are listed below.

1. It is not easy to separate and remove CO₂ as greenhouse gas in the suspension.
2. Some bacteria or viruses strongly interact with the hydrophobic materials as PDMS. In that case, the diffusion effect by H⁺ and HCO₃⁻ for fouling might not work.
3. Can this method separate the high molecular weight fouling materials such as humic acid?

Response to the reviewers' comments

We greatly appreciate all of the time and effort that the Reviewers put into reviewing our paper, and we were pleased to receive many constructive comments as well as several helpful questions to consider. We have carefully considered the feedback from the Reviewers and incorporated their suggestions to improve the quality of our paper. We believe that the detailed responses that follow address all of the questions and comments raised by the Reviewers. We believe that these revisions prompted by the Reviewers' feedback have strengthened the manuscript.

Reviewer 1: Comment 1

This is an interesting and well written contribution.

Microfluidic filtration of colloids has been addressed in a number of publications from Montpellier, Aachen and Twente. It is growing field as to study membrane filtration and colloidal filtration phenomena. (Linkhorst, J., Beckmann, T., Go, D., Kuehne, A. J. C., & Wessling, M. (2016). Microfluidic colloid filtration. *Scientific Reports*, 6, 22376–8. <http://doi.org/10.1038/srep22376>)

The use of CO₂ to establish pH gradients in PDMS based chips has been extensively studied, but not cited. de Jong, J., Verheijden, P. W., Lammertink, R. G. H., & Wessling, M. (2008). Generation of local concentration gradients by gas-liquid contacting. *Analytical Chemistry*, 80(9), 3190–3197. <http://doi.org/10.1021/ac7023602>

Our response

We thank the Reviewer for carefully reviewing our paper and we were very pleased that the reviewer considers it interesting and well written.

Though our main claim is not about creating a chemical gradient itself in a PDMS device - rather we show that such gradients of dissolved gas concentration can induce diffusiophoresis - we agree that the paper by de Jong *et al.* shows a nice way to achieve ion gradients via gas dissolution, which could be useful and informative to the readers. In this regard, we have cited the paper in the main text (Ref. 27).

Reviewer 1: Comment 2

A membrane free separation process in a microfluidic device using pH gradients is novel and worth publishing. The visually observed separation seems very effective. Yet, it would be great to quantify the sharpness of the separation. In membrane filtration, retention curves are used. how long has the channel to be to have no particles left in the particle-free stream? How clean is the "filtrate"? Any thoughts how the LRV (log reduction value) can be?

Our response

Although the presented device is a prototype device that operates at a lab scale, the particle removal rate is shown to be competitive. Owing to the low flow rate (2 $\mu\text{L/hr}$) and high fluorescence intensity of the particles, we were able to count the number of individual particles passing through the filtrate stream (e.g. Fig. R1). During 5 minutes, out of $\sim 2.2 \times 10^7$ particles only 104 pass through the filtrate stream, which corresponds to $\text{LRV}=5.3$.

To clarify this point, we have added the LRV analysis in the paper (and the SI) as follows:

End of paragraph 3 in page 3: *"The particle removal rate in terms of log reduction value (LRV) is estimated to be 5.3, which is comparable to the conventional microfiltration and ultrafiltration techniques [28]."*

[28] F. I. Hai, T. Riley, S. Shawkat, S. F. Magram, and K. Yamamoto, "Removal of pathogens by membrane bioreactors: A review of the mechanisms, influencing factors and reduction in chemical disinfectant dosing," *Water* **6**, 3603-3630 (2014).

Fig. R1. Image sequence of particle (0.5 μm polystyrene) removal driven by CO_2 dissolution. The time between each frame is 0.2 s, which allows tracking of individual particles (indicated by arrows) flowing through the filtrate stream.

Reviewer 1: Comment 3

While the work is conceptually inspiring, it will have a hard time to be implemented. Anybody who has worked with these chips knows how difficult it is to get them operate properly. Filtering a real world solution will clog the device immediately.

Our response

The current study aims at demonstrating a membraneless route to achieve particle removal, which is shown to be energy-efficient. Employing the device for a practical application is still immature at the moment and requires future studies. Nevertheless, we emphasize that our device does not require any porous filter, which effectively prevents fouling induced by steric effects. Also, there are certainly cases where relatively dilute solutions are to be handled and particles need to be removed (e.g., ultrapure water production).

Reviewer 1: Comment 4

Energy consumption is used as an argument why this device is superior of a membrane device. Energy consumption is not much of an issue in dilute streams. Short-term fouling control and long term drift though is. Energy consumption only becomes relevant in high fouling streams operating at high Re numbers due to recirculating up to 90% of the retentate. Yet, such streams cannot be handled by this device.

Our response

We have not made energy consumption the main claim of our paper. However, we have simply noted that this approach is relatively energy efficient, which we think is important for readers to know. The most important aspect of our device is that it is free from cake formation and pore plugging, which are the major contributors to membrane fouling in micro/ultrafiltration techniques [R1]. A possible mechanism for fouling in our device is particle accumulation on the channel surface. However, this is unfavorable in our device since the surface is always exposed to shear due to the continuous flow, thus effectively suppressing the growth of the particle layer.

[R1] X. Zhu and M. Elimelech, "Colloidal fouling of reverse osmosis membranes: Measurements and fouling mechanisms," *Environ. Sci. Technol.* **31**, 3654-3662 (1997).

Reviewer 1: Comment 5

Nonetheless, this paper should be published and the authors may change some of their wording and reflections if they are susceptible to the arguments and their own experience in chip operation.

Our response

Again, we thank the Reviewer for reviewing our paper. We have revised the main text to address the Reviewer's suggestions.

Reviewer 2: Comment 1

The authors address the idea of membrane filtration using CO₂. The core idea is related to field flow fractionation (FFF, as Giddings described), in which particles are pulled toward or away from one region of the capillary cross section, in this case by diffusiophoresis, and then the output is split. The diffusiophoresis is caused by an electrolyte gradient of dissociated carbonic acid, which arises as the CO₂ permeates one side of the PDMS material and dissociates into H⁺ and HCO₃⁻. This manuscript is very interesting, and could be suitable for Nature Comm, if the authors can address the questions and comments below.

- questions. The scientific or engineering questions need to be clarified. Right now the manuscript is not driven by strong questions, but rather by a clever idea.

Our response

We thank the reviewer for the careful review our manuscript and appreciate the feedback that the manuscript is very interesting. We agree that the process may be viewed as an example of FFF, with the novelty being in the way that we apply a force on the particles via a soluble gas that can be removed easily after the separation is achieved. We now mention this in the manuscript, citing Giddings et al. [Ref. 24, 25] (page 3, line 26). Giddings et al. attempted FFF driven by a concentration field, but they were unsuccessful [Ref. 25].

The main scientific question we address is whether the transient ion concentration gradients produced during CO₂ dissolution are sufficient to drive appreciable particle motion via diffusiophoresis, and so effect separations. The answer to this question is yes, which we show in Figure 1 and the associated discussion. The key engineering questions lie in the design of a device that exploits this phenomenon for continuous particle filtration. As we note, further engineering questions remain to be explored to optimize the device to reduce its energy consumption and enhance the extent of separation.

Reviewer 2: Comment 2

Zeta potentials. What is the zeta potential of the amine particles? The negative PSL particles (and what are the surface groups)? The PDMS (and if small, does this allow particle adhesion)? The zetas will vary as a function of pH; the authors have chosen the pH for pN₂ = 136 kPa. Diffusiophoresis depends upon the difference in the zeta potentials of the particles and the surface, and so these are critical to provide in the manuscript.

Our response

The measured zeta potential values of polystyrene and amine-coated polystyrene are -70 mV and +60 mV, respectively. These numbers are provided in the main text (third paragraph in the Results section) and the Supplementary Information. Although zeta potential is known to be dependent on pH, the zeta potential of polystyrene, however, does not change significantly with pH above 3 (Fig. R2

[R2]).

Fig. R2. pH dependence of zeta potential of latex particles. PSt refers to native polystyrene particles whereas PStm and PStn refer to functionalized polystyrenes [R2].

When measuring the zeta potentials, the pH of the particle suspension was at equilibrium with the atmosphere, where $p\text{CO}_2=40$ Pa (pH~5.8). The zeta potential of the surface is only important when diffusioosmosis plays a significant role, which we can neglect in our experiments due to the confined channel geometries.

[2] H. Ohshima and K. Furusawa, *Electrical Phenomena at Interfaces*, 2nd ed., CRC Press, 1998

Reviewer 2: Comment 3

Flow profile. If the zetaPDMS is not 0, then the walls will contribute to the flow profile within the capillary. Have the authors verified the flow profile by examining the particle speeds throughout the depth of the cell (~60 μm). How does this affect the separation? This might be especially important when the surface is not PDMS, if other materials would be used in commercial applications?

Our response

In our study, we have two different experimental setups, which use UV-curable epoxy and PDMS. The epoxy channel (depth ~ 60 μm) is used to demonstrate a well-controlled particle diffusiophoresis driven by CO_2 dissolution whereas the PDMS channel (depth ~ 20 μm) is used for filtration experiments. In both experiments, diffusioosmosis can be neglected since a diffusioosmotic flow induced by chemical gradients will be balanced by back pressure, resulting in a circulating flow which effectively makes the net fluid flow zero at any cross section of the channel [R3,R4]. Furthermore, the average flow speed in the filtration experiments (~300 $\mu\text{m/s}$) is significantly larger than a typical diffusioosmotic flow speed

($O(10 \mu\text{m/s})$). Therefore, the overall particle dynamics is not influenced by the presence of diffusioosmosis.

[R3] A. Kar, T.-Y. Chiang, I. O. Rivera, A. Sen, and D. Velegol, “Enhanced transport into and out of dead-end pores,” *ACS Nano* **9**, 746–753 (2015).

[R4] S. Shin, E. Um, B. Sabass, J. T. Ault, M. Rahimi, P. B. Warren, and H. A. Stone, “Size-dependent control of colloid transport via solute gradients in dead-end channels,” *Proc. Natl. Acad. Sci. U.S.A.* **113**, 257–261 (2016).

Reviewer 2: Comment 4

pN₂. The authors should discuss briefly in the manuscript why they chose pN₂ = 136 kPa. In the SI the authors mention that this is a pressure where the speed of the particles does not change much with pN₂, if I understand correctly? They might explain this in a sentence or two, and offer a reason why this is a good criterion for the choice of pressure?

Our response

We believe the reviewer is referring to the chosen CO₂ pressure rather than the N₂ pressure. The choice of pressure was motivated by practical considerations: it must be slightly higher (in this case we used 5 psi, 34 kPa) than the ambient pressure to flow through the device. The experiments with N₂ used the same pressure as the experiments with CO₂ as a control. The content in the SI shows that the extent of particle motion is not highly sensitive to the applied CO₂ pressure for pressures near atmospheric. This means that there is little benefit to increasing the applied pressure significantly and a significant decrease would be needed to reduce the motion. We now mention the reason for the chosen CO₂ and N₂ pressures in the Methods section as follows:

Page 5, line 38: *“The gas pressure of 136 kPa (5 psi above atmospheric pressure) was chosen to allow flow through the device to an outlet at atmospheric pressure.”*

Reviewer 2: Comment 5

Fouling. On p4 the authors say that the membrane filtration idea is “easily scalable”. What is the role of fouling - due to either particles or bacteria - of the PDMS, especially over months? Would this or other challenges affect the scalability? Or the membrane’s cleaning cycle or useful lifetime?

Our response

One advantage of our device in terms of scalability is that the devices are placed in parallel, allowing independent operation regardless of failure in adjacent channels due to various causes including fouling. And as emphasized in the previous response, our device does not require any porous membrane, which

effectively prevents fouling due to cake formation or pore plugging. In this regard, our device also does not require any back-washing, which is heavily required in membrane-based filtration techniques.

Reviewer 2: Comment 6

Ionic strength. Many application will require finite salt concentrations, perhaps of NaCl or other salt. DP transport is reduced at higher ionic strength. Will the mechanism still be effective, say at 1 mM, 10 mM, 150 mM? What will need to change?

Our response

The effect of finite salt concentrations is an important question for practical applications and requires future studies. Though at this stage we have demonstrated the process in more ideal conditions, we can provide estimates of the effects of finite salt concentrations by solving the Nernst-Planck equations for multispecies transport numerically. In the presence of various background NaCl concentrations, the diffusion potentials established across a 100 μm -wide channel at one second after the exposure to CO_2 ($p\text{CO}_2=136$ kPa) are presented in Table R1.

Table R1: Diffusion potential due to CO_2 dissolution at various background NaCl concentrations

NaCl concentration (mM)	Diffusion potential (mV)
0	27.2
1	1.61
10	0.16
150	0.01

Due to the presence of ionic solutes in the particle suspensions we use, we have added a section to the supplementary information discussing the expected effects of additional ionic solutes. Though reduced significantly, motion is still expected when the background salt concentration exceeds the applied H^+ concentration by 40% (in the SI, this is the case with 0.25% NaN_3 , for which we have 0.2 mM vs. 0.14 mM). One way to achieve separation despite higher ionic strength would be to use higher CO_2 pressures, but this approach is eventually limited by the ion concentration scaling with the square root of the CO_2 pressure. With 1 MPa of CO_2 pressure, the concentration of H^+ reaches 0.4 mM. CO_2 is unlikely to be effective at higher background ion concentrations and a different approach, perhaps the use of a more soluble gas, would be needed to reach 10 or 100 mM ion concentrations.

Reviewer 2: Comment 7

Advantages. The authors mention a 1000x reduction in energy (p 4, paragraph 1), which is a clear advantage. Even a 10x reduction is huge. They should address this with a bit of additional explanation, since this reduction seems so big. It is also very useful that the particle transport depends primarily on the particle zeta potential, and less so the particle size. This can be used to advantage in many situations. Also, the gas transport can be very rapid, which would allow rapid changes in the ionic strength as needed. The authors might point out these additional advantages.

Our response

We thank the reviewer for pointing out these additional advantages. The main reason for the significant reduction in energy dissipation is that the suspension can flow through a much larger channel than the size of the particles that must be removed, which we now point out more clearly in the manuscript as given below. The ease with which gas can be transported into and out of a suspension was one motivation for pursuing this work.

Page 4, line 5: *“Energy dissipation is reduced because the fluid may flow through a channel that is much wider than the particles instead of passing through pores that are smaller than the particles.”*

Reviewer 2: Comment 8

Overall, I do not understand why this paper is suitable for a high impact journal such as Nature Communications. It shows a unique phenomenon and a simple demonstration but I think the overall utility is somewhat oversold.

The comparison of the scalability and the power consumption to current practical membranes is not accurate I think. Current membranes are relatively energy efficient and for simple situations such as those demonstrated in this paper can even be run under gravity. Also the recovery of such membranes is over 95% not 50%. The comparison and scale up from a small microfluidic device to a large industrialized system is not appropriate I think.

Our response

The 50% recovery rate is based on our current device where the outlet is divided into two identical streams. This split ratio is one aspect of the device that may be optimized. Based on Fig. R3, most of the particles are accumulated within 20% of the channel width, which suggests a possibility for further improvement in the recovery rate by reducing the width of the retentate stream. Moreover, the recovery rate can be further enhanced by recirculating the retentate through multiple stages. Such optimization and system-level design for industrial applications requires further effort, which is left for future studies.

Fig. R3. (a) Fluorescence microscopy image of particles (0.5 μm polystyrene) migrating transverse to the flow direction (Fig. 4c in the main text). (b) Intensity distribution along line a–b in (a).

Reviewer 2: Comment 9

Also, membranes provide a barrier for particle filtration, an important consideration due to the need to remove microorganisms, something that the proposed system cannot assure.

Other than this, the work conducted is elegant and provides a nice insight into unusual situations where diffusiophoresis can be utilized.

Our response

Given that most of the microorganisms also possess considerable surface charge [R5], the proposed method is expected to work also for removing microorganisms, which we are planning to pursue in the future. We thank the reviewer once again for the reviewing our paper. We are pleased to hear Reviewer’s conclusion that our work is “elegant and provides a nice insight.”

[R5] W. W. Wilson, M. M. Wade, S. C. Holman, and F. R. Champlin, “Status of methods for assessing bacterial cell surface charge properties based on zeta potential measurements,” *J. Microbiol. Methods* **43**, 153–164 (2001)

Reviewer 3: Comment 1

This manuscript entitled “Membraneless water filtration using CO₂” reports the directed motion of particles induced by exposure to CO₂. The authors demonstrated a scalable, continuous flow, membraneless particle filtration process. The reviewer thinks that the idea is noteworthy and very promising method to clean water and to realize low energy consumption. However, it is difficult to make a judgment whether the water filtration using CO₂ really work for chemical or biological fouling. It is true that the particle migration observed in this manuscript is mainly induced by the diffusion potential due to H⁺ and HCO₃⁻. However, natural organic matter in water has usually been known as complex mixtures consisting of hydrophobic and hydrophilic materials with a broad range of molecular weights.

The reviewer thinks that this manuscript needs additional description for above and the following comments to publish in Nature Communications.

For the authors, comments are listed below.

It is not easy to separate and remove CO₂ as greenhouse gas in the suspension.

Our response

We thank the reviewer for the review of our manuscript and the positive feedback that the idea is noteworthy and promising. Natural water sources contain a variety of electrolyte and nonelectrolyte solutes, of which electrolytes would interfere with the operating mechanism of the process we propose. Consequently, the process is restricted to applications in which the water entering has a lower ionic strength than that of a solution saturated with CO₂ at the applied pressure.

Ionic solutes are present in the particle suspensions we use, and our results show the effectiveness of the method despite their presence. As detailed in the revised supplementary information, we estimate that motion, though significantly reduced, is still expected when the background salt concentration exceeds the applied H⁺ concentration by 40% (the case with 0.25% NaN₃, for which we have 0.2 mM vs. 0.14 mM). One way to achieve motion despite higher ionic strength would be to use higher CO₂ pressures, but this approach is eventually limited by the ion concentration scaling with the square root of the CO₂ pressure. With 1 MPa of CO₂ pressure, the concentration of H⁺ reaches 0.4 mM. CO₂ is unlikely to be effective at higher background ion concentrations and a different approach, perhaps the use of a more soluble gas, would be needed to reach 10 or 100 mM ion concentrations.

The addition and removal of added CO₂ from a suspension is easy in a PDMS microfluidic device, and the use of other gases is also possible. We now cite Jong *et al.* (Ref. 27) in support of this claim.

Reviewer 3: Comment 2

Some bacteria or viruses strongly interact with the hydrophobic materials as PDMS. In that case, the

diffusion effect by H⁺ and HCO₃ for fouling might not work.

Our response

The surface chemistry of PDMS can be tuned by various chemical methods [R6], which may also mitigate adhesion of bacteria onto the channel surface. As an example, the PDMS channel was coated with amine (Fig. 4d,e in the main text) to prevent adhesion of positively-charged particles onto the channel surface, which will otherwise result in rapid adhesion of the particles.

[R6] J. Zhou, A. V. Ellis, and N. H. Voelcker, “Recent developments in PDMS surface modification for microfluidic devices,” *Electrophoresis* **31**, 2-16 (2010).

Reviewer 3: Comment 3

Can this method separate the high molecular weight fouling materials such as humic acid?

Our response

We thank the reviewer for drawing our attention to humic acids due to their relevance to treating water from natural sources. At present we have chosen to focus on using the process for separating macroscopic particles. However, it is likely that ion concentration gradients due to gas dissolution can be used to induce migration of other ionic species, and futures studies with humic acids would be valuable to pursue.

REVIEWERS' COMMENTS:

Reviewer #2 (Remarks to the Author):

I thank the authors for revising their manuscript. They have addressed the original concerns. Still the manuscript is driven by a clever idea, more than an interesting fundamental question. Perhaps that is OK for this case. Their main claim is that gradients of dissolved gas concentrations can produce DP. This is interesting, although not greatly surprising scientifically.

More interesting is their technology, which appears to have the possibility of dramatically altering membrane separations for many cases. While the separation ability is very good, they have not shown key factors in scaling up can be overcome: 1) high throughput (and this is where the idea might be most important for energy reduction), 2) medium to high salt concentrations, or 3) the presence of fouling proteins or bacterial EPS. I think readers will want to have at least some idea of how to address these factors.

Two more small points:

- The authors state that they can neglect the DO in their system due to the "confined channel geometries". Wouldn't confined channels make DO even more important, since DO is efficient in thin capillaries or channels? Later the authors stated that the average flow speed in the expts (~300 $\mu\text{m/s}$) make pressure-driven convection dominant. This latter explanation makes more sense.

- I don't see why the CO₂ conc need be as high as the background conc? The background conc will be uniform, while the ions from the dissolved CO₂ will cause DP. The background electrolyte might reduce the DP effect, but will not stop it. You will just get a slower DP transport; you might simply need a longer channel, much like FFF or HDC.

Reviewer #3 (Remarks to the Author):

Since the reviewer received the constructive answers from the authors, the reviewer thinks that this revised manuscript should be published in Nature Communication.

Response to the reviewers' comments

We greatly appreciate the time and effort that the reviewers put into reviewing our revised manuscript, and we were pleased to receive their constructive comments and helpful questions. We have carefully considered the feedback from the reviewers and incorporated their suggestions to improve the quality of our paper. We believe that the detailed responses that follow address all of the questions and comments raised by the reviewers. These revisions prompted by the reviewers' feedback have strengthened the manuscript.

Reviewer 2

I thank the authors for revising their manuscript. They have addressed the original concerns. Still the manuscript is driven by a clever idea, more than an interesting fundamental question. Perhaps that is OK for this case. Their main claim is that gradients of dissolved gas concentrations can produce DP. This is interesting, although not greatly surprising scientifically.

More interesting is their technology, which appears to have the possibility of dramatically altering membrane separations for many cases. While the separation ability is very good, they have not shown key factors in scaling up can be overcome: 1) high throughput (and this is where the idea might be most important for energy reduction), 2) medium to high salt concentrations, or 3) the presence of fouling proteins or bacterial EPS. I think readers will want to have at least some idea of how to address these factors.

Our response

The strategy for scaling up requires careful optimization to determine the dimensions of the channels, the flow rates, and the number of channels. Frictional (pressure) losses can be reduced by using a wider or taller (in the direction of the concentration gradient) channel, but increasing the channel height reduces the magnitude of the concentration gradient. Higher flow rates (decreased residence times) and reduced gradients can be compensated by using a longer channel, at the expense of increasing the required energy input. The optimal choice is not obvious and requires further experimental and theoretical analysis, as we now mention in more detail in the discussion section.

Application of the process to suspensions with appreciable salt concentrations also presents an optimization question. The reduction in the driving force with increasing salt concentration could be compensated by decreasing the size of the flow channel (to increase the concentration gradient) or lengthening the device, both of which increase energy requirements.

A systematic study of the effects of increasing salt concentrations is straightforward though time consuming, and likely requires consideration of a variety of ionic species representative of those found in typical water sources (sodium and chloride, but also potassium, magnesium, calcium, carbonate, and sulfate, etc, which all have different diffusivities). The most reasonable way to determine the effects of

fouling proteins or bacterial extracellular polymeric substances is through long-term experiments with representative water samples.

Reviewer 2

Two more small points:

- The authors state that they can neglect the DO in their system due to the "confined channel geometries". Wouldn't confined channels make DO even more important, since DO is efficient in thin capillaries or channels? Later the authors stated that the average flow speed in the expts (~300 $\mu\text{m/s}$) make pressure-driven convection dominant. This latter explanation makes more sense.

Our response

We had used the term “confined geometries” to describe a system that is closed such that the geometry is effectively a dead-end. As the reviewer has mentioned, DO flow can be effective at transporting particles in thin capillaries, but only if the channel has outlets. Otherwise, DO flow will create a circulating motion whose net flow is zero at any given cross section. To clarify, we have added the following statement in the Supplementary Discussion (page 4):

We neglect diffusioosmosis due to the wall surface charge since a Poiseuille flow driven by back pressure cancels the diffusioosmotic flow, making a zero net fluid flow in any given cross-section of the channel. Not only is this approximation valid for the stationary experimental case in Fig. 2, but it is also true for the continuous flow filtration device in Fig. 4 because the ion gradients are established in the direction transverse to the main flow direction. Furthermore, the large speed of the main flow ($\approx 300 \mu\text{m/s}$) compared to diffusioosmotic flow, $O(10 \mu\text{m/s})$, would allow one to neglect the diffusioosmotic flow when predicting the particle motion in the filtration devices.

Reviewer 2

- I don't see why the CO_2 conc need be as high as the background conc? The background conc will be uniform, while the ions from the dissolved CO_2 will cause DP. The background electrolyte might reduce the DP effect, but will not stop it. You will just get a slower DP transport; you might simply need a longer channel, much like FFF or HDC.

Our response

We thank the reviewer for this observation and suggestion. A CO_2 concentration that generates an ion

concentration exceeding the background ion concentration is not a necessary condition for separating particles via diffusiophoresis, but this is advantageous since it ensures a high driving force. The diffusiophoretic driving force is coupled with the flux of background ions, whose concentration distributions would be altered by the added ions, and it decreases with increasing concentration due to the logarithmic dependence of the particle speed on the ion concentration. One can increase the length of the channel or decrease its height to compensate for a reduced driving force, and this leads to the optimization problem discussed previously. Eventually, the magnitude of diffusiophoretic motion becomes similar to the Brownian motion of the particles, making difusiophoretic separation ineffective. An in-depth analysis of how the displacement of particles depends on the background ion concentration requires further theoretical (likely computational) and experimental work. We have summarized the main points mentioned here in an additional paragraph in the discussion section of the manuscript.

Reviewer 3

Since the reviewer received the constructive answers from the authors, the reviewer thinks that this revised manuscript should be published in Nature Communication.

Our response

We thank Reviewer 3 for their time and effort in reviewing the manuscript.